# Reduced antibody cross-reactivity following infection with B.1.1.7 than with parental SARS-CoV-2 strains

Nikhil Faulkner[1,2†], Kevin W Ng[1†], Mary Y Wu[3†], Ruth Harvey[4†], Marios Margaritis[5], Stavroula Paraskevopoulou[5], Catherine Houlihan[5,6], Saira Hussain[4,7], Maria Greco[7], William Bolland[1], Scott Warchal[3], Judith Heaney[5], Hannah Rickman[5], Moria Spyer[5,8], Daniel Frampton[6], Matthew Byott[5], Tulio de Oliveira[9,10,11,12], Alex Sigal[9,13,14], Svend Kjaer[15], Charles Swanton[16], Sonia Gandhi[17], Rupert Beale[18], Steve J Gamblin[19], John W McCauley[4], Rodney Stuart Daniels[4], Michael Howell[3], David Bauer[7], Eleni Nastouli[1,5,8], George Kassiotis[1,20]*

[1]Retroviral Immunology, London, United Kingdom; [2]National Heart and Lung Institute, Imperial College London, London, United Kingdom; [3]High Throughput Screening STP, London, United Kingdom; [4]Worldwide Influenza Centre, London, United Kingdom; [5]Advanced Pathogen Diagnostics Unit UCLH NHS Trust, London, United Kingdom; [6]Division of Infection and Immunity, London, United Kingdom; [7]RNA Virus Replication Laboratory, London, United Kingdom; [8]Department of Population, Policy and Practice, London, United Kingdom; [9]School of Laboratory Medicine and Medical Sciences, University of KwaZulu-Natal, Durban, South Africa; [10]KwaZulu-Natal Research Innovation and Sequencing Platform, Durban, South Africa; [11]Centre for the AIDS Programme of Research in South Africa, Durban, South Africa; [12]Department of Global Health, University of Washington, Seattle, United States; [13]Africa Health Research Institute, Durban, South Africa; [14]Max Planck Institute for Infection Biology, Berlin, Germany; [15]Structural Biology STP, London, United Kingdom; [16]Cancer Evolution and Genome Instability Laboratory, London, United Kingdom; [17]Neurodegradation Biology Laboratory, London, United Kingdom; [18]Cell Biology of Infection Laboratory, London, United Kingdom; [19]Structural Biology of Disease Processes Laboratory, The Francis Crick Institute, London, United Kingdom; [20]Department of Infectious Disease, St Mary's Hospital, Imperial College London, London, United Kingdom

*For correspondence:
george.kassiotis@crick.ac.uk

†These authors contributed equally to this work

Competing interests: The authors declare that no competing interests exist.

## Abstract

**Background:** The degree of heterotypic immunity induced by severe acute respiratory syndrome coronavirus 2 (SARS-CoV-2) strains is a major determinant of the spread of emerging variants and the success of vaccination campaigns, but remains incompletely understood.

**Methods:** We examined the immunogenicity of SARS-CoV-2 variant B.1.1.7 (Alpha) that arose in the United Kingdom and spread globally. We determined titres of spike glycoprotein-binding antibodies and authentic virus neutralising antibodies induced by B.1.1.7 infection to infer homotypic and heterotypic immunity.

**Results:** Antibodies elicited by B.1.1.7 infection exhibited significantly reduced recognition and neutralisation of parental strains or of the South Africa variant B.1.351 (Beta) than of the infecting variant. The drop in cross-reactivity was significantly more pronounced following B.1.1.7 than parental strain infection.

**Conclusions:** The results indicate that heterotypic immunity induced by SARS-CoV-2 variants is asymmetric.

**Funding:** This work was supported by the Francis Crick Institute and the Max Planck Institute for Dynamics of Complex Technical Systems, Magdeburg.

## Introduction

Mutations in severe acute respiratory syndrome coronavirus 2 (SARS-CoV-2) variants that arose in the United Kingdom (UK) (B.1.1.7; Alpha) or in South Africa (B.1.351; Beta) reduce recognition by antibodies elicited by natural infection with the parental reference (Wuhan) strain and the subsequent D614G variant (*Cele et al., 2021*; *Diamond et al., 2021*; *Edara et al., 2021*; *Emary et al., 2021*; *Liu et al., 2021b*; *Planas et al., 2021*; *Skelly et al., 2021*; *Wang et al., 2021*; *Wibmer et al., 2021*; *Zhou et al., 2021*). Such reduction in cross-reactivity also impinges the effectiveness of current vaccines based on the Wuhan strain (*Diamond et al., 2021*; *Edara et al., 2021*; *Emary et al., 2021*; *Liu et al., 2021b*; *Skelly et al., 2021*; *Wang et al., 2021*; *Zhou et al., 2021*), prompting consideration of alternative vaccines based on the new variants. However, the immunogenicity of the latter or, indeed, the degree of heterotypic immunity the new variants may afford remains to be established.

## Results and Discussion

The B.1.1.7 variant is thought to have first emerged in the UK in September 2020 and has since been detected in over 50 countries (*Kirby, 2021*). To examine the antibody response to B.1.1.7, we collected sera from 29 patients, admitted to University College London Hospitals (UCLH) for unrelated reasons (*Supplementary file 1*), who had confirmed B.1.1.7 infection. The majority (23/29) of these patients displayed relatively mild COVID-19 symptoms and a smaller number (6/29) remained COVID-19-asymptomatic. As antibody titres may depend on the severity of SARS-CoV-2 infection, as well as on time since infection (*Gaebler et al., 2021*; *Long et al., 2020*), we compared B.1.1.7 sera with sera collected during the first wave of D614G variant spread in London from hospitalised COVID-19 patients (*Ng et al., 2020*) (n=20) and mild/asymptomatic SARS-CoV-2-infected health care workers (*Houlihan et al., 2020*) (n=17) who were additionally sampled 2 months later.

IgG, IgM, and IgA antibodies to the spikes of the Wuhan strain or of variants D614G, B.1.1.7, or B.1.351, expressed on HEK293T cells, were detected by a flow cytometry-based method (*Figure 1*; *Figure 1—figure supplement 1*; *Ng et al., 2020*). Titres of antibodies that bound the parental D614G spike largely correlated with those that bound the B.1.1.7 or B.1.351 spikes (*Figure 1a–c*), consistent with the high degree of similarity. Similar correlations were observed for all three Ig classes also between the Wuhan strain and the three variant spikes and between the B.1.1.7 and B.1.351 spikes (*Figure 1—figure supplements 2–5*).

Comparison of sera from acute D614G and B.1.1.7 infections revealed stronger recognition of the infecting variant than of other variants. Although B.1.1.7 sera were collected on average earlier than D614G sera (*Supplementary file 1*), titres of antibodies that bound the homotypic spike or neutralised the homotypic virus, as well as the relation between these two properties, were similar in D614G and B.1.1.7 sera (*Figure 1—figure supplement 6a–c*), suggesting comparable immunogenicity of the two variants. Moreover, levels of binding and neutralising antibodies were not statistically significantly different in sera from mild or asymptomatic B.1.1.7 infection, although they were, on average, lower in the latter (*Figure 1—figure supplement 6d*).

Recognition of heterotypic spikes was reduced by a small, but statistically significant degree for both D614G and B.1.1.7 sera and for all three Ig classes (*Figure 1d–f*). IgM or IgA antibodies in both D614G and B.1.1.7 sera were less cross-reactive than IgG antibodies (*Figure 1d–f*). The direction of cross-reactivity was disproportionally affected for some combinations, with IgA antibodies in D614G sera retaining on average 81% of recognition of the B.1.1.7 spike and IgA antibodies in B.1.1.7 sera retaining on average 30% of recognition of the D614G spike (*Figure 1f*). Similarly, recognition of the B.1.351 spike by IgM antibodies was retained, on average, to 71% in D614G sera and to 46% in B.1.1.7 sera (*Figure 1f*). Measurable reduction in polyclonal antibody binding to heterotypic spikes was unexpected, given >98% amino acid identity between them. Furthermore,

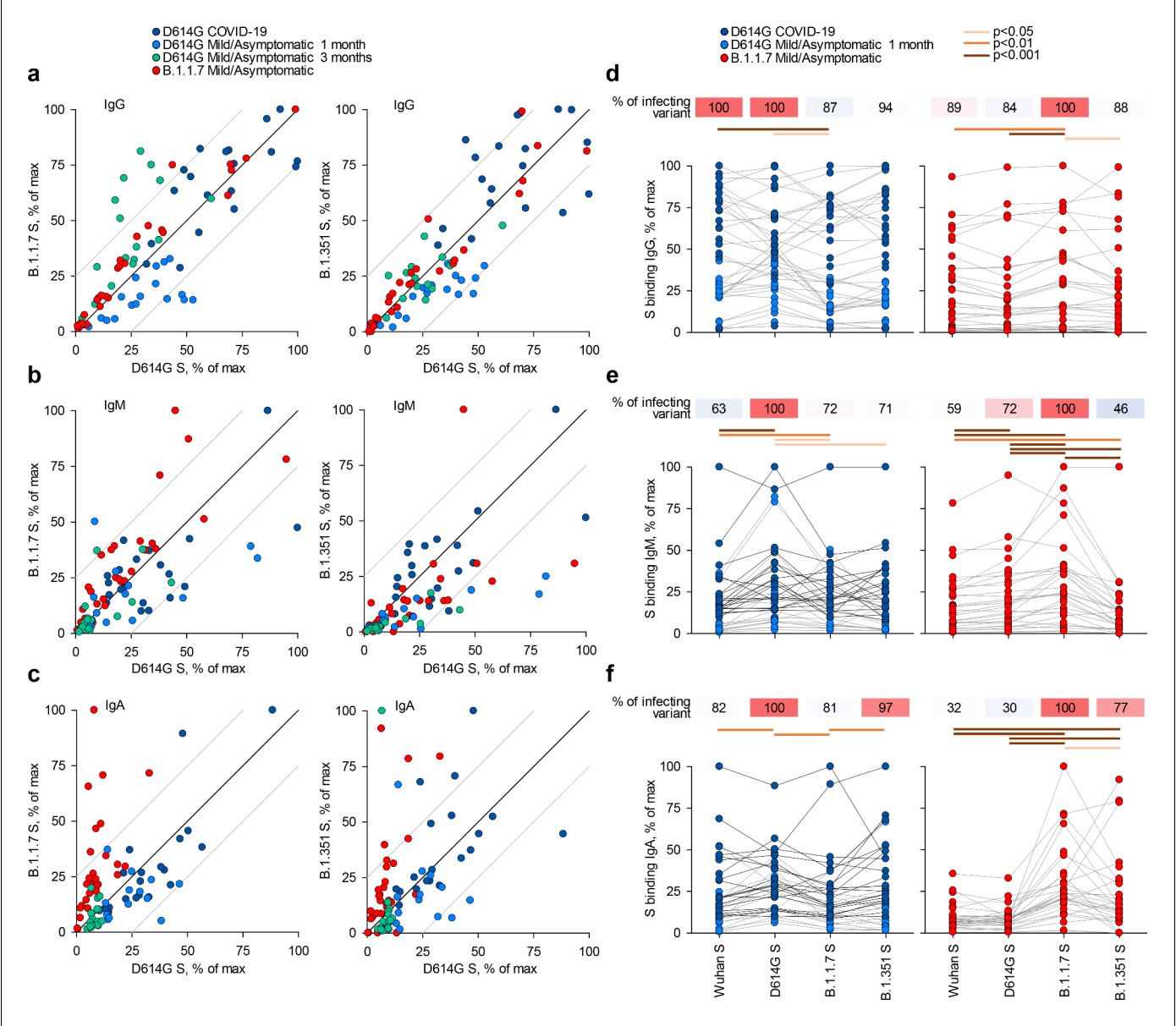

**Figure 1.** Recognition of distinct severe acute respiratory syndrome coronavirus 2 (SARS-CoV-2) spike glycoproteins by antibodies in D614G and B.1.1.7 sera. (a–c) Correlation of IgG (a), IgM (b), and IgA (c) antibody levels to D614G and B.1.1.7 or B.1.351 spikes in the indicated groups of donors infected either with the D614G or B.1.1.7 strains. Each symbol represents an individual sample and levels are expressed as a percentage of the positive control. Black lines denote complete correlation and grey lines a 25% change in either direction. (d–f) Comparison of IgG (d), IgM (e), and IgA (f) antibody levels to the indicated spikes in groups of donors acutely infected either with the D614G or B.1.1.7 strains. Connected symbols represent individual donors. Numbers above the plots denote the average binding to each spike, expressed as a percentage of binding to the infecting spike.

The online version of this article includes the following figure supplement(s) for figure 1:

**Figure supplement 1.** Flow cytometric detection of spike-binding antibodies.

**Figure supplement 2.** Recognition of distinct severe acute respiratory syndrome coronavirus 2 (SARS-CoV-2) spike glycoproteins by antibodies in D614G and B.1.1.7 sera.

**Figure supplement 3.** Recognition of distinct severe acute respiratory syndrome coronavirus 2 (SARS-CoV-2) spike glycoproteins by antibodies in D614G and B.1.1.7 sera.

**Figure supplement 4.** Recognition of distinct severe acute respiratory syndrome coronavirus 2 (SARS-CoV-2) spike glycoproteins by antibodies in D614G and B.1.1.7 sera.

**Figure supplement 5.** Matrix of correlation coefficients between binding and neutralising antibodies.

**Figure supplement 6.** Kinetics and magnitude of the antibody response to D614G and B.1.1.7 infection.

mutations selected for escape from neutralising antibodies, which target the receptor binding domain more frequently, should not directly affect binding of non-neutralising antibodies to other domains of the spike. Indeed, we found that the reduction in heterotypic binding was less pronounced than the reduction in heterotypic neutralisation. However, reduction in serum antibody binding has also been observed for the receptor binding domain of the B.1.351 spike (*Edara et al., 2021*). Together, these findings suggested that either the limited number of mutated epitopes were targeted by a substantial fraction of the response (*Diamond et al., 2021*; *Skelly et al., 2021*; *Wang et al., 2021*; *Zhou et al., 2021*) or allosteric effects or conformational changes affecting a larger fraction of polyclonal antibodies.

To examine a functional consequence of reduced antibody recognition, we measured the half maximal inhibitory concentration ($IC_{50}$) of D614G and B.1.1.7 sera using in vitro neutralisation of authentic Wuhan or B.1.1.7 and B.1.351 viral isolates (*Figure 2a–b*). Titres of neutralising antibodies correlated most closely with levels of IgG binding antibodies for each variant (*Figure 1—figure supplement 5*). Neutralisation of B.1.1.7 by D614G sera was largely preserved at levels similar to neutralisation of the parental Wuhan strain (fold change −1.3; range 3.0 to −3.8, p=0.183) (*Figure 2b*), consistent with other recent reports, where authentic virus neutralisation was tested (*Brown et al., 2021*; *Diamond et al., 2021*; *Planas et al., 2021*; *Skelly et al., 2021*; *Wang et al., 2021*). Thus, D614G infection appeared to induce substantial cross-neutralisation of the B.1.1.7 variant. However, the reverse was not true. Neutralisation of the parental Wuhan strain by B.1.1.7 sera was significantly reduced, compared to neutralisation of the infecting B.1.1.7 variant (fold change −3.4; range −1.20 to −10.6, p<0.001) (*Figure 2b*), and the difference in cross-neutralisation drop was also significant

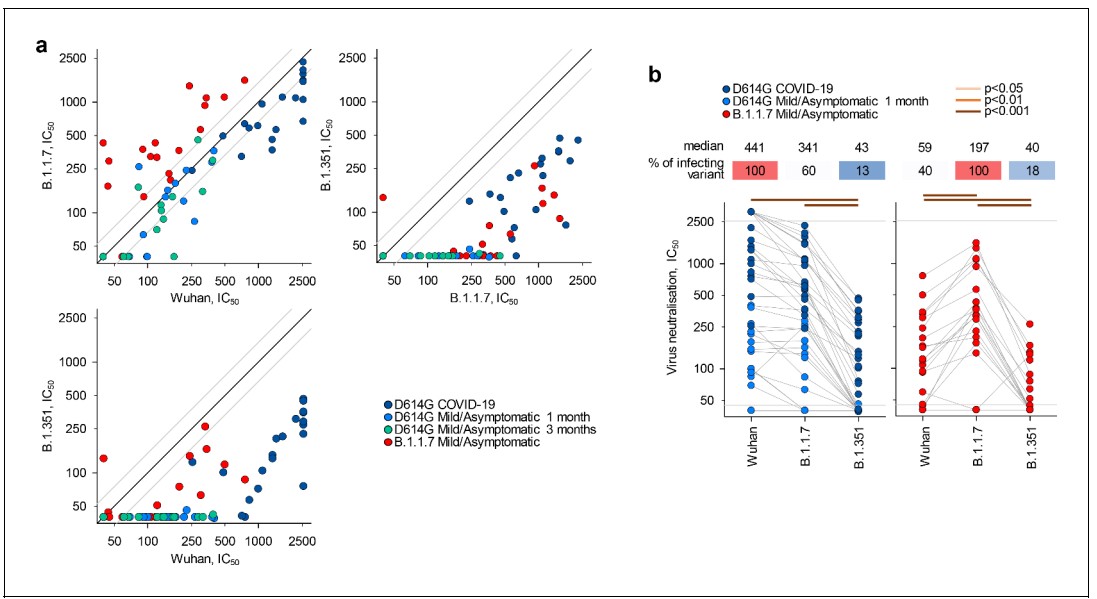

**Figure 2.** Neutralisation of distinct severe acute respiratory syndrome coronavirus 2 (SARS-CoV-2) strains by antibodies in D614G and B.1.1.7 sera. (**a**) Correlation of neutralising antibody levels ($IC_{50}$) against the Wuhan, B.1.1.7, or B.1.351 strains in the indicated groups of donors infected either with the D614G or B.1.1.7 strains. Each symbol represents an individual sample. Black lines denote complete correlation and grey lines a 50% (twofold) change in either direction. (**b**) Comparison of neutralising antibody levels ($IC_{50}$) to the indicated SARS-CoV-2 strains in groups of donors acutely infected with either the D614G or B.1.1.7 strains. Connected symbols represent individual donors. Numbers above the plots denote the average $IC_{50}$ against each strain, expressed as a percentage of $IC_{50}$ against the infecting strain. Grey horizontal lines denote the lower and upper limit of detection.

The online version of this article includes the following figure supplement(s) for figure 2:

**Figure supplement 1.** Neutralisation of distinct severe acute respiratory syndrome coronavirus 2 (SARS-CoV-2) strains by antibodies in D614G sera, according to severity of infection.

**Figure supplement 2.** Binding and neutralising antibodies at a 3-month follow-up of mild/asymptomatic D614G infection.

**Figure supplement 3.** Neutralisation of distinct severe acute respiratory syndrome coronavirus 2 (SARS-CoV-2) strains by antibodies in D614G and B.1.1.7 sera from mild/asymptomatic infection.

**Figure supplement 4.** Spike sequence distance of severe acute respiratory syndrome coronavirus 2 (SARS-CoV-2) variants.

**Figure supplement 5.** Severe acute respiratory syndrome coronavirus 2 (SARS-CoV-2) neutralisation assay setup.

(p<0.001). Both D614G and B.1.1.7 sera displayed significantly reduced neutralisation of the B.1.351 variant with a fold change of −8.2 (range −1.7 to −33.5) and −7.7 (range −3.4 to −17.9), respectively (*Figure 2b*).

Although B.1.1.7 infection appeared to induce limited heterotypic immunity, relative to D614G infection, differences in both the severity of infection with each variant and the time since infection may have affected the degree of antibody cross-reactivity observed. For example, higher SARS-CoV-2-neutralising antibody titres are found in infections leading to severe COVID-19 than in mild/asymptomatic infection (*Long et al., 2020*) and these higher titres may include broader antibody diversity. Similarly, a longer time since infection may permit broader antibody diversity through somatic hypermutation and affinity maturation (*Gaebler et al., 2021*), potentially increasing cross-reactivity. However, the stronger heterotypic recognition of B.1.1.7 by D614G sera was independent of severity of infection and was, in fact, more pronounced in mild/asymptomatic than in severe D614G infection, when the two were considered separately, with sera from severe and mild/asymptomatic D614G infection retaining 52% and 85% neutralisation of B.1.1.7 (*Figure 2—figure supplement 1*). Moreover, the ability of sera from mild/asymptomatic D614G to neutralise B.1.1.7 did not change over time (*Figure 2—figure supplement 2*). Indeed, whilst binding antibody titres were significantly reduced for all three Ig classes in D614G sera in the 2 months of follow-up, neutralising antibody titres remained comparable for the Wuhan and B.1.1.7 strains and were undetectable at both time-points for the B.1.351 strain (*Figure 2—figure supplement 2*). Lastly, to adjust for potentially confounding differences in both the severity of infection and time since infection with each variant, we compared a subset of 11 seropositive samples from D614G or B.1.1.7 infection. These were selected for comparable disease outcome (all mild/asymptomatic) and for time since confirmed infection (on average, 24.0 and 19.5 days, respectively, p=0.37). Analysis of these comparable subsets further supported the notion that B.1.1.7 infection elicited reduced heterotypic immunity, with D614G and B.1.1.7 sera retaining 87% and 42% neutralisation of B.1.1.7 and D614G, respectively, and much lower neutralisation of B.1.351 (*Figure 2—figure supplement 3*).

Together, these results argue that natural infection with each SARS-CoV-2 strain induces antibodies that recognise the infecting strain most strongly, with variable degrees of cross-recognition of the other strains. Importantly, antibodies induced by B.1.1.7 infection were less cross-reactive with other dominant SARS-CoV-2 strains than those induced by the parental strain. Similar findings were recently obtained independently by Brown et al., who found that B.1.1.7 convalescent sera neutralised the parental strain significantly less than the infecting B.1.1.7 strain (*Brown et al., 2021*). Conversely, sera from D614G infection retained full neutralisation of the B.1.1.7 strain (*Brown et al., 2021*). This unidirectional pattern of cross-reactivity argues that emergence of B.1.1.7 is unlikely to have been driven by antibody escape. In support of this premise, B.1.1.7 and D614G viruses were equally sensitive to neutralisation by BNT162b2 or AZD1222 vaccination-induced antibodies, although they were both approximately twofold less sensitive than the Wuhan strain (*Wall et al., 2021a*; *Wall et al., 2021b*).

In contrast to the results reported here and by Brown et al., Liu et al. recently reported that B.1.1.7 convalescent sera recognised significantly stronger the Victoria strain (a Wuhan related strain) than homotypic B.1.1.7 virus, and retained stronger heterotypic recognition of other variants of concern (VOCs) than sera from infection with D614G, B.1.351, or with variant B.1.1.28 (Gamma) first emerged in Brazil (*Liu et al., 2021a*). Methodological differences notwithstanding, it is possible that donor selection may be responsible for the reported differences in antibody levels and cross-reactivity. Of note, neutralising antibody titres in B.1.1.7 sera were two to three times higher than those in sera from any other infection in Liu et al., suggesting higher immunogenicity of the B.1.1.7 infection compared with all other strains (*Liu et al., 2021a*). In contrast, overall antibody titres induced by B.1.1.7 infection were comparable with those induced by parental strain infection in this study (*Figure 1—figure supplement 6a–c*) and in Brown et al., when tested against the homotypic strains (*Brown et al., 2021*). Nevertheless, it is possible that the higher viral loads achieved during B.1.1.7 infection than D614G infection (*Frampton et al., 2021*) also induce higher antibody levels in B.1.1.7 sera than in D614G sera. Consequently, even though, relative to recognition of the infecting strain, B.1.1.7 sera may be less cross-reactive than D614G sera, they may still harbour higher antibody titres than D614G sera against other strains in absolute terms. Indeed, our comparison of B.1.1.7 and D614G sera from donors we attempted to match for severity and time of serum collection since infection indicated that B.1.1.7 sera contained higher absolute levels of neutralising antibodies than

D614G sera against the infecting variant (p=0.003) and against the B.1.351 variant (p=0.006). Although analysis of larger numbers of samples will be required to conclusively determine if B.1.1.7 infection is more immunogenic than D614G infection, the current data highlight the effect of the severity of infection on resulting antibody titres and the importance of controlling for such confounding factors.

In addition to the emergence and global spread of the B.1.1.7 variant, several other variants have emerged, such as variant B.1.617.2 (Delta), first emerged in India, that has now replaced variant B.1.1.7 in the UK. Assessment of the extent of heterotypic immunity induced by new variants will be critical for understanding of the degree of infection-induced immunity against other variants and for adapting current vaccines. A recent comparison of sera from infection with B.1.351 or the parental strain B.1.1.117 in South Africa also observed stronger neutralisation of the infecting strain (*Cele et al., 2021*). In contrast to B.1.1.7 infection, however, B.1.351 infection induced substantial cross-neutralisation of the parental strain, as well as of the B.1.1.28 variant, whereas parental strain B.1.1.117 infection induced significantly lower B.1.351 neutralisation (*Cele et al., 2021*; *Moyo-Gwete et al., 2021*). Therefore, heterotypic immunity in the case of B.1.351 and the parental strain B.1.1.117 was also asymmetrical, but reversed.

The B.1.351, B.1.1.28, and B.1.617.2 VOCs appear comparably sensitive to antibodies induced by the BNT162b2 and AZD1222 vaccines, which are both based on the Wuhan sequence (*Liu et al., 2021a*; *Wall et al., 2021a*; *Wall et al., 2021b*). However, infection with the B.1.351 or the B.1.1.28 variant may induce lower cross-neutralisation of the other variant than itself (*Liu et al., 2021a*), likely owing to spike sequence divergence between them (*Figure 2—figure supplement 4*). The cross-reactivity of antibodies induced by B.1.617.2 infection is currently unknown, but spike sequence divergence considerations would predict an even lower degree of heterotypic immunity. Indeed, whereas the spike proteins of all current VOCs harbour between 10 and 12 amino acid changes from the Wuhan reference spike sequence, they harbour between 12 and 21 amino acid changes between them, with B.1.617.2 being the most divergent at present (*Figure 2—figure supplement 4*). It stands to reason that the more divergent their spike sequences become, the lower the degree of heterotypic immunity the variants induce. This degree of heterotypic immunity should be an important consideration in the choice of spike variants as vaccine candidates. The antigenic variation associated with SARS-CoV-2 evolution may instead necessitate the use of multivalent vaccines.

## Materials and methods

### Key resources table

| Reagent type (species) or resource | Designation | Source or reference | Identifiers | Additional information |
|---|---|---|---|---|
| Antibody | BV421 anti-human IgG (monoclonal) | Biolegend | RRID:AB_2562176; Cat# 409318 | FACS (1:200) |
| Antibody | APC anti-human IgM (monoclonal) | Biolegend | RRID:AB_493011; Cat# 314510 | FACS (1:200) |
| Antibody | PE anti-human IgA (monoclonal) | Miltenyi Biotech | RRID:AB_2733860; Cat# 130-114-002 | FACS (1:200) |
| Antibody | Anti-SARS-CoV-2 S2 clone D001 (monoclonal) | SinoBiological | RRID:AB_2857932; Cat# 40590-D001 | FACS |
| Antibody | Alexa488 anti-SARS-CoV-2 nucleoprotein (monoclonal) | Produced in-house | CR3009 | IF |
| Recombinant DNA reagent | pcDNA3-SARS-CoV-2_WT spike | Dr Massimo Pizzato, University of Trento, Italy | Wuhan spike sequence | Transfected construct |
| Recombinant DNA reagent | pcDNA3-SARS-CoV-2_D614G spike | Dr Massimo Pizzato, University of Trento, Italy | Wuhan spike sequence with D614G mutation and cytoplasmic tail deletion | Transfected construct |

*Continued on next page*

*Continued*

| Reagent type (species) or resource | Designation | Source or reference | Identifiers | Additional information |
|---|---|---|---|---|
| Recombinant DNA reagent | pcDNA3-SARS-CoV-2_B.1.1.7 spike | This paper | B.1.1.7 spike sequence | Transfected construct |
| Recombinant DNA reagent | pcDNA3-SARS-CoV-2_ B.1.351 spike | This paper | B.1.351 spike sequence | Transfected construct |
| Cell line (*Homo sapiens*) | HEK293T | Cell Services facility at the Francis Crick Institute | RRID:CVCL_0063; CVCL_0063 | |
| Cell line (*Chlorocebus* sp.) | Vero E6 | Dr Björn Meyer, Institut Pasteur, Paris, France | CRL-1586 | |
| Cell line (*Chlorocebus* sp.) | Vero V1 | Prof. Steve Goodbourn, St. George's, University of London, London, UK | CCL-81 | |
| Other | SARS-CoV-2 | hCoV-19/England/02/2020 | Respiratory Virus Unit, Public Health England, UK | Wuhan strain |
| Other | SARS-CoV-2 | hCoV-19/England/204690005/2020 | Public Health England (PHE), UK, through Prof. Wendy Barclay, Imperial College London, London, UK | B.1.1.7 strain |
| Other | SARS-CoV-2 | 501Y.V2.HV001 *Cele et al., 2021* | | B.1.351 strain |

## Donor and patient samples and clinical data

Serum or plasma samples from D614G infection were obtained from UCLH (REC ref: 20/HRA/2505) COVID-19 patients (n=20, acute D614G infection, COVID-19 patients) as previously described (*Ng et al., 2020*), or from UCLH health care workers (n=17, acute D614G infection, mild/asymptomatic), as previously described (*Houlihan et al., 2020*; *Supplementary file 1*). These samples were collected between March 2020 and June 2020. Serum or plasma samples from B.1.1.7 infection were obtained from patients (n=29, acute B.1.1.7 infection, mild/asymptomatic) admitted to UCLH (REC ref: 20/HRA/2505) for unrelated reasons, between December 2020 and January 2021, who then tested positive for SARS-CoV-2 infection by RT-qPCR, as part of routine testing (*Supplementary file 1*). Infection with B.1.1.7 was confirmed by sequencing of viral RNA, covered from nasopharyngeal swabs. A majority of these patients (n=23) subsequently developed mild COVID-19 symptoms and six remained asymptomatic. All serum or plasma samples were heat-treated at 56℃ for 30 min prior to testing. No statistical methods were used to compute sample size for a pre-determined effect size. All patients/participants who had consented and were available at the time of the study were included.

## Diagnosis of SARS-CoV-2 infection by RT-qPCR and next-generation sequencing

SARS-CoV-2 nucleic acids were detected in nasopharyngeal swabs from hospitalised patients by a diagnostic RT-qPCR assay using custom primers and probes (*Grant et al., 2020*). Assays were run by Health Services Laboratories (HSL), London, UK. Diagnostic RT-qPCR assays for SARS-CoV-2 infection in health care workers was run at the Francis Crick Institute, as previously described (*Aitken et al., 2020*). SARS-CoV-2 RNA-positive samples (RNA amplified by Aptima Hologic) were subjected to real-time whole-genome sequencing at the UCLH Advanced Pathogen Diagnostics Unit. RNA was extracted from nasopharyngeal swab samples on the QiaSymphony platform using the Virus Pathogen Mini Kit (Qiagen). Libraries were prepared using the Illumina DNA Flex library preparation kit and sequenced on an Illumina MiSeq (V2) using the ARTIC protocol for targeted amplification (primer set V3). Genomes were assembled using an in-house pipeline (*ICONIC Consortium et al., 2017*) and aligned to a selection of publicly available SARS-CoV-2

genomes (*Elbe and Buckland-Merrett, 2017*) using the MAFFT alignment software (*Katoh and Standley, 2013*). Phylogenetic trees were generated from multiple sequence alignments using IQ-TREE (*Nguyen et al., 2015*) and FigTree (http://tree.bio.ed.ac.uk/software/figtree), with lineages assigned (including B.1.1.7 calls) using pangolin (http://github.com/cov-lineages/pangolin), and confirmed by manual inspection of alignments.

## Cells lines and plasmids

HEK293T cells were obtained from the Cell Services facility at the Francis Crick Institute, verified as mycoplasma-free and validated by DNA fingerprinting. Vero E6 and Vero V1 cells were kindly provided by Dr Björn Meyer, Institut Pasteur, Paris, France, and Prof. Steve Goodbourn, St. George's, University of London, London, UK, respectively. Cells were grown in Iscove's Modified Dulbecco's Medium (Sigma-Aldrich) supplemented with 5% fetal bovine serum (Thermo Fisher Scientific), L-glutamine (2 mM, Thermo Fisher Scientific), penicillin (100 U/ml, Thermo Fisher Scientific), and streptomycin (0.1 mg/ml, Thermo Fisher Scientific). For SARS-CoV-2 spike expression, HEK293T cells were transfected with an expression vector (pcDNA3) carrying a codon-optimised gene encoding the wild-type full-length SARS-CoV-2 reference spike (referred to here as Wuhan spike, UniProt ID: P0DTC2) or a variant carrying the D614G mutation and a deletion of the last 19 amino acids of the cytoplasmic tail (referred to here as D614G spike) (both kindly provided by Massimo Pizzato, University of Trento, Italy). Similarly, HEK293T cells were transfected with expression plasmids (pcDNA3) encoding the full-length B.1.1.7 spike variant (D614G, Δ69–70, Δ144, N501Y, A570D, P681H, T716I, S982A, and D1118H) or the full-length B.1.351 spike variant (D614G, L18F, D80A, D215G, L242H, R246I, K417N, E484K, N501Y, A701V) (both synthesised and cloned by GenScript). All transfections were carried out using GeneJuice (EMD Millipore) and transfection efficiency was between 20% and 54% in separate experiments.

## SARS-CoV-2 isolates

The SARS-CoV-2 reference isolate (referred to as the Wuhan strain) was the hCoV-19/England/02/2020, obtained from the Respiratory Virus Unit, Public Health England, UK (GISAID EpiCov accession EPI_ISL_407073). The B.1.1.7 isolate was the hCoV-19/England/204690005/2020, which carries the D614G, Δ69–70, Δ144, N501Y, A570D, P681H, T716I, S982A, and D1118H mutations (*Brown et al., 2021*; *Figure 2—figure supplement 4*), obtained from Public Health England (PHE), UK, through Prof. Wendy Barclay, Imperial College London, London, UK. The B.1.351 virus isolate was the 501Y.V2.HV001, which carries the D614G, L18F, D80A, D215G, Δ242–244, K417N, E484K, N501Y, A701V mutations (*Cele et al., 2021*; *Figure 2—figure supplement 4*). However, sequencing of viral genomes isolated following further passage in Vero V1 cells identified the Q677H and R682W mutations at the furin cleavage site, in approximately 50% of the genomes. All viral isolates were propagated in Vero V1 cells.

## Flow cytometric detection of antibodies to spike glycoproteins

HEK293T cells were transfected to express the different SARS-CoV-2 spike variants. Two days after transfection, cells were trypsinised and transferred into V-bottom 96-well plates (20,000 cells/well). Cells were incubated with sera (diluted 1:50 in PBS) for 30 min, washed with FACS buffer (PBS, 5% BSA, 0.05% sodium azide), and stained with BV421 anti-IgG (clone HP6017, Biolegend), APC anti-IgM (clone MHM-88, Biolegend), and PE anti-IgA (clone IS11-8E10, Miltenyi Biotech) for 30 min (all antibodies diluted 1:200 in FACS buffer). Expression of SARS-CoV-2 spike was confirmed by staining with the D001 antibody (40590-D001, SinoBiological). Cells were washed with FACS buffer and fixed for 20 min in CellFIX buffer (BD Bioscience). Samples were run on a Ze5 analyzer (Bio-Rad) running Bio-Rad Everest software v2.4 or an LSR Fortessa with a high-throughput sampler (BD Biosciences) running BD FACSDiva software v8.0, and analyzed using FlowJo v10 (Tree Star Inc) analysis software, as previously described (*Ng et al., 2020*). All runs included three positive control samples, which were used for normalisation of mean fluorescence intensity (MFI) values. To this end, the MFI of the positively stained cells in each sample was expressed as a percentage of the MFI of the positive control on the same 96-well plate. The results shown are from one of one to two independent experiments.

## SARS-CoV-2 neutralisation assay

SARS-CoV-2 variant neutralisation was tested using an in-house developed method (*Figure 2—figure supplement 5*). Heat-inactivated serum samples in QR coded vials (FluidX/Brooks) were assembled into 96-well racks along with foetal calf serum-containing vials as negative controls and SARS-CoV-2 spike RBD-binding nanobody (produced in-house) vials as positive controls. A Viaflo automatic pipettor fitted with a 96-channel head (Integra) was used to transfer serum samples into V-bottom 96-well plates (Thermo 249946) prefilled with Dulbecco's modified eagle medium to achieve a 1:10 dilution. The Viaflo was then used to serially dilute from the first dilution plate into three further plates at 1:4 to achieve 1:40, 1:160, and 1:640. Next, the diluted serum plates were stamped into duplicate 384-well imaging plates (Greiner 781091) pre-seeded the day before with 3000 Vero E6 cells per well, with each of the four dilutions into a different quadrant of the final assay plates to achieve a final working dilution of samples at 1:40, 1:160, 1:640, and 1:2560. Assay plates were then transferred to containment level 3 (CL3) where cells were infected with the indicated SARS-CoV-2 viral strain, by adding a pre-determined dilution of the virus prep using a Viaflo fitted with a 384 head with tips for the no-virus wells removed. Plates were incubated for 24 hr at 37°C, 5% $CO_2$ and then fixed by adding a concentrated formaldehyde solution to achieve a final concentration of 4%. Assay plates were then transferred out of CL3 and fixing solution washed off, cells blocked, and permeabilised with a 3% BSA/0.2% Triton-X100/PBS solution, and finally immunostained with DAPI and an Alexa488-conjugated anti-nucleoprotein monoclonal antibody (clone CR3009; produced in-house). Automated imaging was carried out using an Opera Phenix (Perkin Elmer) with a 5× lens and the ratio of infected area (Alexa488-positive region) to cell area (DAPI-positive region) per well calculated by the Phenix-associated software Harmony. A custom automated script runs plate normalisation by background subtracting the median of the no-virus wells and then dividing by the median of the virus-only wells before using a three-parameter dose-response model for curve fitting and identification of the dilution which achieves 50% neutralisation for that particular serum sample ($IC_{50}$). The results shown are from one of two to three independent experiments.

## Statistical analyses

Data were analysed and plotted in SigmaPlot v14.0 (Systat Software). Parametric comparisons of normally distributed values that satisfied the variance criteria were made by paired or unpaired Student's t-tests or one-way analysis of variance tests. Data that did not pass the variance test were compared with Wilcoxon signed rank tests.

## Acknowledgements

We are grateful for assistance from the Flow Cytometry and Cell Services facilities at the Francis Crick Institute and to Mr Michael Bennet and Mr Simon Caidan for training and support in the high-containment laboratory. We wish to thank the Public Health England (PHE) Virology Consortium and PHE field staff, the ATACCC (Assessment of Transmission and Contagiousness of COVID-19 in Contacts) investigators, the G2P-UK (Genotype to Phenotype-UK) National Virology Consortium, and Prof. Wendy Barclay, Imperial College London, London, UK, for the B.1.1.7 viral isolate. This work was supported by the Francis Crick Institute, which receives its core funding from Cancer Research UK, the UK Medical Research Council, and the Wellcome Trust. The funders had no role in study design, data collection and analysis, decision to publish, or preparation of the manuscript.

## Additional information

### Funding

| Funder | Author |
| --- | --- |
| Francis Crick Institute | Nikhil Faulkner |
| | Kevin W Ng |
| | Mary Y Wu |
| | Ruth Harvey |
| | Saira Hussain |
| | Maria Greco |
| | William Bolland |

|  | Scott Warchal |
|  | Svend Kjaer |
|  | Charles Swanton |
|  | Sonia Gandhi |
|  | Rupert Beale |
|  | Steve j Gamblin |
|  | John W McCauley |
|  | Rodney Stuart Daniels |
|  | Michael Howell |
|  | David Bauer |
|  | George Kassiotis |
| Max Planck Institute for Dynamics of Complex Technical Systems Magdeburg | Alex Sigal |

The funders had no role in study design, data collection and interpretation, or the decision to submit the work for publication.

## Author contributions

Nikhil Faulkner, Kevin W Ng, Mary Y Wu, Ruth Harvey, Saira Hussain, Maria Greco, William Bolland, Scott Warchal, Formal analysis, Investigation; Marios Margaritis, Stavroula Paraskevopoulou, Catherine Houlihan, Judith Heaney, Hannah Rickman, Moria Spyer, Daniel Frampton, Matthew Byott, Tulio de Oliveira, Alex Sigal, Svend Kjaer, Rupert Beale, Resources; Charles Swanton, Sonia Gandhi, Steve J Gamblin, John W McCauley, Rodney Stuart Daniels, Michael Howell, David Bauer, Eleni Nastouli, Supervision; George Kassiotis, Conceptualization, Supervision, Writing - original draft

## Author ORCIDs

Kevin W Ng  http://orcid.org/0000-0003-1635-6768
Mary Y Wu  https://orcid.org/0000-0002-2074-6171
Alex Sigal  http://orcid.org/0000-0001-8571-2004
Svend Kjaer  http://orcid.org/0000-0001-9767-8683
Rupert Beale  http://orcid.org/0000-0002-6705-8560
John W McCauley  http://orcid.org/0000-0002-4744-6347
George Kassiotis  https://orcid.org/0000-0002-8457-2633

## Ethics

Human subjects: Serum or plasma samples were obtained from University College London Hospitals (UCLH) (REC ref: 20/HRA/2505).

## Decision letter and Author response

Decision letter https://doi.org/10.7554/eLife.69317.sa1
Author response https://doi.org/10.7554/eLife.69317.sa2

## Additional files

### Supplementary files

- Source data 1. Binding and neutralising titre values.

- Supplementary file 1. Donor and patient characteristics. This table lists the number, median age (and range), gender proportion, and the median time (and range) post infection for the donors and patients studied.

- Transparent reporting form

### Data availability

All data generated or analysed during this study are included in the manuscript and supporting files.

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
