## [Decision Letter]

**Acceptance summary:**

This study describes reduced antibody cross-reactivity between the SARS-CoV-2 B.1.1.7 variant and the parental strain or the B.1.351 variant. Asymmetric antibody responses and reduced neutralizing antibodies against heterogeneous variants have been demonstrated in multiple studies. The current study reports reduction of B.1.1.7 COVID-19 sera against the SARS-CoV-2 parental strain and B.1.351. This observation is interesting and could be useful for future vaccine development. The work is of interests to virologists and infectious disease specialists.

**Decision letter after peer review:**

Thank you for submitting your article "Reduced antibody cross-reactivity following infection with B.1.1.7 than with parental SARS-CoV-2 strains" for consideration by eLife. Your article has been reviewed by 3 peer reviewers, one of whom is a member of our Board of Reviewing Editors, and the evaluation has been overseen by Bavesh Kana as the Senior Editor. The reviewers have opted to remain anonymous.

Essential Revisions:

1. As the authors are aware, the B.1.617.2 VOC has become highly topical and is currently circulating around the world. Recent studies have shown similar results for B.1.1.7 and also B.1.617.2 (for example, https://www.cell.com/cell/fulltext/S0092-8674(21)00755-8). These other studies should be cited and discussed in the manuscript to better contextualise this study's findings.

2. In sup. Figure S9 the authors present their most controlled (like-for-like) comparison of sera from WT and B.1.1.7 infected individuals. They conclude that B.1.1.7 infection leads to nAbs with lower cross reactivity. But this is only true for the fold drop. Looking at the absolute level of nAbs in these two groups, it seems to indicate the opposite conclusion is plausible. That is, if one gets an asymptomatic B.1.1.7 infection, it seems they may have a higher neutralisation against B.1.1.7 and WT (and even B.1.351) compared with someone who receives an asymptomatic WT infection. This would seem to indicate that a B.1.1.7 infection leads to a higher overall response including a higher cross-reactive response. This interpretation is the opposite of the authors, and the difference relies on whether absolute level or fold-drop is counted as the better measure of cross-reactivity. It may be that B.1.1.7 infections tend to be asymptomatic with higher viral loads and so lead to higher overall Ab responses with asymptomatic infection. It is not clear which of these two opposite interpretations of the data is the "correct" interpretation and the authors approach is very reasonable one – it's just not clear which is the most meaningful at this stage. Therefore, the authors should include some discussion that another interpretation is possible when looking at the absolute level of nAbs in B.1.1.7 infected individuals and offer a balanced justification for why they favour the interpretation that B.1.1.7 leads to lower cross-reactivity instead of higher cross reactivity.

3. Do the expressed B.1.1.7 and B.1.351 spikes have the cytoplasmic tail removed?

Editorial/clarification:

1. Figure 1a and Figure 1c should have x axis labelled.

2. A schematic or figure showing the mutations for each variant might be useful to see how different they are from each other.

3. Currently available vaccines are based on the Wuhan sequence and sera collected in this study are from D614G infected patients. Can the authors comment on the effects D614G has on antigenicity?

4. It would be worth showing the median IC50s along with % of infecting variant. For example, in Figure S7 the IC50s of sera from severe D614G infection are still high despite a 50% decrease from the infecting variant.

4. It is interesting that IgG binding is largely retained across all variant spikes yet neutralisation drops significantly against B.1.351. Do the authors have any explanations for this?

5. If asymptomatic patients are considered, are there any differences with binding or neutralisation when compared to mild?

6. Typo on page 4 – seconds last line "IgG biding antibodies".

*Reviewer #3 (Recommendations for the authors):*

I have several questions and comments.

1. Do the expressed B.1.1.7 and B.1.351 spikes have the cytoplasmic tail removed?

2. Figure 1a and Figure 1c should have x axis labelled.

3. A schematic or figure showing the mutations for each variant might be useful to see how different they are from each other.

4. Currently available vaccines are based on the Wuhan sequence and sera collected in this study are from D614G infected patients. Can the authors comment on the effects D614G has on antigenicity?

5. I think it would be worth showing the median IC50s along with % of infecting variant. For example, in Figure S7 the IC50s of sera from severe D614G infection are still high despite a 50% decrease from the infecting variant.

6. It is interesting that IgG binding is largely retained across all variant spikes yet neutralisation drops significantly against B.1.351. Do the authors have any explanations for this?

7. If you look at asymptomatic patients are there any differences with binding or neutralisation when compared to mild?

8. Is there a reason D614G mild/asymptomatic 3 month data is not included in some of the figures?

9. Typo on page 4 – seconds last line "IgG biding antibodies".

---

## [Author Response]

Essential Revisions (for the authors):1. As the authors are aware, the B.1.617.2 VOC has become highly topical and is currently circulating around the world. Recent studies have shown similar results for B.1.1.7 and also B.1.617.2 (for example, https://www.cell.com/cell/fulltext/S0092-8674(21)00755-8). These other studies should be cited and discussed in the manuscript to better contextualise this study's findings.

Testament to how quickly SARS-CoV-2 (and our understanding of it) evolves, since submission of this work, B.1.617.2 started spreading in the UK, replacing B.1.1.7, and we and others have since reported the reduced sensitivity of this variant to vaccine-induced antibodies (Wall et al., 2021a, Wall et al., 2021b in the revised manuscript). The B.1.617.2 variant was not available when the experiments reported here were performed.

We have now discussed the new variant and new studies published during review of this work. The consensus is that each variant induces the strongest response to itself and the more the variants diverge, the lower the heterotypic immunity they induce. There are, however, notable differences too. For example, the new study from Oxford (Liu et al., 2021a) uniquely finds that B.1.1.7 sera recognise better the Victoria strain (a Wuhan related strain) than B.1.1.7 itself and these differences, as well as the implications for vaccine adaptation, have now been discussed.

2. In sup. Figure S9 the authors present their most controlled (like-for-like) comparison of sera from WT and B.1.1.7 infected individuals. They conclude that B.1.1.7 infection leads to nAbs with lower cross reactivity. But this is only true for the fold drop. Looking at the absolute level of nAbs in these two groups, it seems to indicate the opposite conclusion is plausible. That is, if one gets an asymptomatic B.1.1.7 infection, it seems they may have a higher neutralisation against B.1.1.7 and WT (and even B.1.351) compared with someone who receives an asymptomatic WT infection. This would seem to indicate that a B.1.1.7 infection leads to a higher overall response including a higher cross-reactive response. This interpretation is the opposite of the authors, and the difference relies on whether absolute level or fold-drop is counted as the better measure of cross-reactivity. It may be that B.1.1.7 infections tend to be asymptomatic with higher viral loads and so lead to higher overall Ab responses with asymptomatic infection. It is not clear which of these two opposite interpretations of the data is the "correct" interpretation and the authors approach is very reasonable one – it's just not clear which is the most meaningful at this stage. Therefore, the authors should include some discussion that another interpretation is possible when looking at the absolute level of nAbs in B.1.1.7 infected individuals and offer a balanced justification for why they favour the interpretation that B.1.1.7 leads to lower cross-reactivity instead of higher cross reactivity.

We thank the Reviewer for raising this important issue. Despite our best efforts to match all other variables in the comparison of B.1.1.7 and parental strain convalescent sera in our collection, confounding differences still remain. We, therefore, compared each the fold drop of the response to other variants, only in relation to the response to the infecting variant in each donor (internally controlled), which is independent of all other variables. We conclude that, for a given response to the infecting variant, relative recognition of parental strains is lower for B.1.1.7 sera than of B.1.1.7 strain for parental strain sera.

Nevertheless, we fully agree with the Reviewer that, given the enormous variability in the antibody response to any variant between individuals, cross-reactivity to another variant may well be higher in absolute terms for at least some B.1.1.7 sera than for parental strain sera, even if it’s lower in relative terms. Also, in light of our previous findings (Frampton et al., 2021) that, on average, B.1.1.7 infection results in increased viral loads than parental strain infection, it could be argued that heterotypic immunity following B.1.1.7 infection will be, on average, higher than following parental strain infection in absolute terms. The discussion has now been revised to reflect these comments.

3. Do the expressed B.1.1.7 and B.1.351 spikes have the cytoplasmic tail removed?

With the exception of the D614G spike, all other spike variants were full-length and contained the intact cytoplasmic domain. This has now been clarified in the Methods section.

Editorial/clarification:1. Figure 1a and Figure 1c should have x axis labelled.

Labels have now been added to all x and y axes in Figure 1a-c. For consistency, the x-axis label has been added also to Figure 2a.

2. A schematic or figure showing the mutations for each variant might be useful to see how different they are from each other.

We have now included a figure (new Figure 2—figure supplement 10) depicting the distance of the spike protein sequences, as well as shared and unique mutations among the variants, and discussed such divergence in the context of cross-reactivity.

3. Currently available vaccines are based on the Wuhan sequence and sera collected in this study are from D614G infected patients. Can the authors comment on the effects D614G has on antigenicity?

During review of this manuscript, we were able to directly compare the effect of the D614G mutation on antigenicity and sensitivity to antibody recognition, using sera also from vaccinated donors. The results of this comparison (Wall et al., 2021a, Wall et al., 2021b) show that the D614G mutation reduces antigenicity in comparison with the Wuhan strain by a factor of ~2.3. However, sensitivity of the D614G and B.1.1.7 variants to vaccinee sera was indistinguishable. These findings are now discussed also in this manuscript.

4. It would be worth showing the median IC50s along with % of infecting variant. For example, in Figure S7 the IC50s of sera from severe D614G infection are still high despite a 50% decrease from the infecting variant.

As per Reviewer’s suggestion, median IC50 values have been added to all the plots.

4. It is interesting that IgG binding is largely retained across all variant spikes yet neutralisation drops significantly against B.1.351. Do the authors have any explanations for this?

As only a fraction of binding antibodies are neutralising, it should be expected that binding is less affected than neutralisation by mutations in emerging variants. There may be little selection pressure to propagate or fix mutations that affect binding of non-neutralising antibodies. In contrast, the selection pressure for mutation that affect neutralisation will be greater in an increasingly immune population. The relatively small number of mutations seen in B.1.351 and other variants of concern (with the exception of the B.1.1.7 variant) appear to be neutralising antibody escape mutations. However, these mutations affect less than 2% of the overall spike sequence and, therefore, binding of antibodies to the unmutated regions of the spike that may not be relevant for neutralisation should be retained. This potential explanation has now been added to the text.

5. If asymptomatic patients are considered, are there any differences with binding or neutralisation when compared to mild?

We have now plotted binding and neutralising antibody levels in B.1.1.7-infected donors according to symptoms (new Figure 1—figure supplement 6d). Although levels of all antibodies were lower in asymptomatic cases than in mild cases, differences were not statistically significant with this small number of asymptomatic cases.

6. Typo on page 4 – seconds last line "IgG biding antibodies".

We thank the Reviewer for spotting this error, which has now been corrected.